# Molecular co-assembled strategy tuning protein conformation for cartilage regeneration

Chengkun Zhao[1,2,8], Xing Li[1,2,8], Xiaowen Han[3], Zhulian Li[1,2], Shaoquan Bian[4], Weinan Zeng[5], Mingming Ding[6], Jie Liang[1,2,7], Qing Jiang[1,2], Zongke Zhou[5], Yujiang Fan[1,2], Xingdong Zhang[1,2] & Yong Sun[1,2] ✉

The assembly of oligopeptide and polypeptide molecules can reconstruct various ordered advanced structures through intermolecular interactions to achieve protein-like biofunction. Here, we develop a "molecular velcro"-inspired peptide and gelatin co-assembly strategy, in which amphiphilic supramolecular tripeptides are attached to the molecular chain of gelatin methacryloyl via intra-/intermolecular interactions. We perform molecular docking and dynamics simulations to demonstrate the feasibility of this strategy and reveal the advanced structural transition of the co-assembled hydrogel, which brings more ordered β-sheet content and 10-fold or more compressive strength improvement. We conduct transcriptome analysis to reveal the role of co-assembled hydrogel in promoting cell proliferation and chondrogenic differentiation. Subcutaneous implantation evaluation confirms considerably reduced inflammatory responses and immunogenicity in comparison with type I collagen. We demonstrate that bone mesenchymal stem cells-laden co-assembled hydrogel can be stably fixed in rabbit knee joint defects by photocuring, which significantly facilitates hyaline cartilage regeneration after three months. This co-assembly strategy provides an approach for developing cartilage regenerative biomaterials.

The development and application of functional injectable protein hydrogels provide a strategy for tissue regeneration under minimally invasive surgery[1]. As a crucial constituent of the extracellular matrix of cartilage, collagen is widely applied as one of the best biomimetic extracellular matrices (ECM) for stem cells in cartilage repair[2]. However, the collagen extracted from animals has some drawbacks, such as insufficient mechanical stability, rapid degradation rate, and potential immunogenicity[3]. Although some collagen derivatives (hydrolyzed proteins, peptides, and recombinant human collagen) have been extensively studied on account of their relatively low level of immunogenicity, issues of insufficient mechanical strength and biological performance have yet to be solved[4,5]. Therefore, it is vital to develop new injectable hydrogels with high strength and satisfactory biological performance to assist cartilage regeneration in situ.

[1]National Engineering Research Center for Biomaterials, Sichuan University, 29# Wangjiang Road, Chengdu, Sichuan 610064, P. R. China. [2]College of Biomedical Engineering, Sichuan University, 29# Wangjiang Road, Chengdu, Sichuan 610064, P. R. China. [3]NHC Key Laboratory of Nuclear Technology Medical Transformation, Mianyang Central Hospital, Mianyang, Sichuan 621099, P. R. China. [4]Shenzhen Institutes of Advanced Technology, Chinese Academy of Sciences, Shenzhen 518055, P. R. China. [5]Department of Orthopedic Surgery and Orthopedic Research Institution, West China Hospital, Sichuan University, Chengdu 610041, China. [6]College of Polymer Science and Engineering, State Key Laboratory of Polymer Materials Engineering, Sichuan University, Chengdu 610065, P. R. China. [7]Sichuan Testing Center for Biomaterials and Medical Devices, Sichuan University, 29# Wangjiang Road, Chengdu 610064, P. R. China. [8]These authors contributed equally: Chengkun Zhao, Xing Li. ✉e-mail: sunyong8702@scu.edu.cn

The structure of proteins is a critical element for determining their mechanical properties and biological roles. Protein structures are generally influenced by their self-assembly behaviors, which mainly rely on inter- and intramolecular interactions[6]. Many basic structural units of living systems, such as peptides and proteins, can self-assemble to form various ordered advanced structures through intermolecular interactions in the external environment to achieve satisfactory protein-like biofunction. With the advancement of supramolecular chemistry, protein or peptide self-assembly behaviors can be regulated at the nanoscale via physical means (e.g., hydrogen bonding, electrostatic force, hydrophobic interactions) or chemical methods (e.g., modification, cross-linking, grafting)[4,7–9]. These strategies have proven to be feasible and have resulted in the development of functional protein hydrogels with superior properties. Consequently, research in this area has increasingly focused on reconstructing active, or even deactivated, proteins by modulating their assembly behavior to improve biofunction.

Gelatin is a collagen derivative with a lower level of immunogenicity, which was on account of the destruction of the immunogenic structure of native collagens[10–13]. The compositional homology ensures a variety of Arg-Gly-Asp (RGD) amino acid sequences in the gelatin chains, which endows it with better biological performance in comparison with synthetic polymers[14]. However, the destruction of the triple helix that occurs during collagen hydrolysis results in incomplete biological performance and unstable physicochemical properties[15]. Although the physicochemical instability could be partly relieved via chemical and physical modification, the high-level protein structure and biological performance cannot be adequately recovered by random cross-linking[16]. The neglect of inter- and intramolecular self-assembly behavior of polypeptides might be one of the key factors. We speculated that the introduction of functional peptide fragments acting on the molecular chain of gelatin, regulating its self-assembly process through the inter- and intramolecular interactions of amino acids, might endow it with specific biofunctions. We previously reported a type of fibrillar supramolecular short peptide (BPAA-GFF), composed of the tripeptide GFF conjugated to 4-biphenylacetic acid (BPAA). The characteristic fibrillar ECM-like structure formed through self-assembly could promote chondrocyte proliferation and maintain its hyaline cartilage phenotype[17,18]. Taking oligopeptide and polypeptide molecules as the research object, we constructed a multi-level and multi-component ordered molecular assembly by using molecular interactions, and studied the assembly mechanism and its role on cartilage regeneration.

In this study, inspired by the physical adhesion of velcro, the BPAA-GFF molecules were employed as "molecular velcro" to attach to the amino acid sequence of gelatin methacryloyl (GelMA) by potential hydrogen bonding among amino acids and hydrophilic/hydrophobic interactions. Meanwhile, methacrylic acid was introduced into gelatin to endow it with in-situ forming and interfacial bonding capabilities, as well as to further enhance the mechanical strength. Molecular docking and molecular dynamics simulation were implemented to confirm the molecular mechanism of the co-assembly hydrogel. The co-assembled fibrous hydrogel promoted stem cell proliferation and differentiation, and specific matrix secretion as shown by transcriptome analysis. BMSCs-laden co-assembled hydrogel regenerated hyaline cartilage in rabbit articular cartilage defects, demonstrating its potential usage in clinical biomaterials for cartilage regeneration (Fig. 1).

## Results

### The construction and optimization of co-assembled hydrogel

First, GelMA with 76.74% substitution degree and BPAA-GFF were prepared[10,17], and the co-assembled hydrogel was achieved (Supplementary Fig. 1, Fig. 2a). The hypothesis that the "molecular velcro" (BPAA-GFF) adhered to GelMA via intermolecular interactions was verified through molecular docking, which predicted the binding affinity by estimating the spatial and energy matching of molecules.

The rough and detailed combination sites observed from molecular docking suggested that the connection between BPAA-GFF and GelMA was driven by intermolecular interactions, especially hydrogen-bonds (H-bonds), Pi-cation interactions, and so on (Fig. 2b–d, Supplementary Fig. 2). In order to optimize the composition and structure of the molecular co-assembled system, we prepared three types of co-assembled hydrogel (Gel-co-LGFF, Gel-co-MGFF, and Gel-co-HGFF) with different BPAA-GFF content. The injectable force and compression modulus were both enhanced with the increased BPAA-GFF content (Fig. 2e). Surprisingly, the compression modulus of Gel-co-HGFF was 14 times as high as that of GelMA (Fig. 2f, Supplementary Fig. 3). These improvements might be attributed to the alteration of intra- or intermolecular interactions and structure transition.

The characteristic peaks of C=O and N-H stretching vibrations exhibited a slight red shift, suggesting H-bond enhancement in co-assembled hydrogels[19,20] (Fig. 2g, Supplementary Fig. 4). The red shift of the C=H characteristic peak in biphenyl also revealed Pi-Pi stacking interactions, which stimulated BPAA-GFF to self-assemble into nanofibers[21] (Fig. 2g). Scanning electron microscope (SEM) and transmission electron microscope (TEM) images showed the highly ordered pore and woven-fiber structure of Gel-co-MGFF (Fig. 2h1–i4). Furthermore, appropriate interactions facilitated the close connection between BPAA-GFF and GelMA in Gel-co-MGFF, which yielded higher stability (Fig. 2j). Meanwhile, these interactions protected molecular co-assembled systems from fast degradation (Fig. 2k). These improved physical and chemical properties made Gel-co-MGFF the most appropriate choice for cell culture and cartilage regeneration. This was supported by cell counting kit-8 (CCK-8) and Live/dead staining results, which revealed that the proliferation efficiency and viability of BMSCs was optimal in Gel-co-MGFF (Fig. 2l, m). Additionally, clearer and more complete cytoskeleton formed a round morphology, suggesting a better chondrogenic differentiation effect[22] (Fig. 2n). Although, the photocuring initiated by LAP under UV light (λ = 405 nm) could potentially affect BMSCs proliferation (Supplementary Fig. 5), Gel-co-MGFF still presented optimal cell proliferation among all the groups.

### Structural transition of the molecular co-assembled hydrogel via intermolecular interactions

To observe the structural alteration of Gel-co-MGFF at the microscopic scale, we studied the structural oscillation of GelMA upon the introduction of BPAA-GFF. During the co-assembly process, the Gel-co-MGFF structure became more compact because BPAA-GFF, acting as "molecular velcro" bound to the inner protein chains and interacted with different regions of GelMA (Fig. 3a, Supplementary Fig. 6). The lower values of the radius of gyration (Rg), root mean square deviation (RMSD), solvent accessible surface area (SASA), and root mean square fluctuation (RMSF) also confirmed the more compact structure and higher stability of Gel-co-MGFF in comparison with GelMA (Fig. 3b–e). To illustrate the reason for optimizations in structure and properties, we compared the interactions of GelMA and Gel-co-MGFF. The improvements in structure and stability were derived from the larger number of intermolecular interactions, such as H-bonds, salt bridges, Pi-Pi stacking, and Pi-cation (Fig. 3f1, f2). Among these interactions, H-bonds played a critical role in the interaction of BPAA-GFF with GelMA, which might be the main driving force of the co-assembly process (Fig. 3f3). Furthermore, the proportions of the secondary structure also changed with increasing numbers of interactions. Compared with GelMA, Gel-co-MGFF showed more β-sheet and less other secondary structure content (Fig. 3g, Supplementary Fig. 7).

These results were also verified through spectroscopy. The characteristic absorption peak of benzene in Gel-co-MGFF exhibited a slight red shift in comparison with GelMA, suggesting more Pi-Pi stacking content. Additionally, the maximum absorption peak ($\lambda_{max}$) was lower in Gel-co-MGFF than in BPAA-GFF, which illustrated the

formation of Pi-Pi stacking between BPAA-GFF and GelMA (Fig. 3h). Thioflavin T (ThT) staining was also employed to observe and investigate the content and distribution of β-sheet[23]. The higher quantity of ThT in Gel-co-MGFF suggested a higher content of β-sheet than in GelMA (Fig. 3i, j). The secondary structure proportion analyzed from the Fourier transform infrared (FT-IR) spectrum also exhibited a similar transition trend (Fig. 3k, l, Table S1). More β-sheet formation was another reason for higher mechanical strength, which might endow Gel-co-MGFF with better biological properties (Fig. 3m).

## Biological mechanism of Gel-co-MGFF regulating the fate of BMSCs

Transcriptomic analysis was employed to investigate the underlying biological mechanism. Pearson correlation analysis (Fig. 4a) indicated good correlation between the biological replicates within two groups. Volcano plots showed 326 upregulated and 221 downregulated genes in a Gel-co-MGFF vs. GelMA comparison (Fig. 4b). The differentially expressed genes were analyzed via Gene Ontology (GO) terms and Kyoto Encyclopedia of Genes and Genomes (KEGG) pathways. For the GO database analysis, the upregulated genes in Gel-co-MGFF vs. GelMA were enriched in cell adhesion, biological adhesion, and cell-cell signaling genes, as well as extracellular region, collagen trimer, and

transporter activity (Fig. 4c). Meanwhile, the downregulated genes were mainly involved in proteolysis, oxidation-reduction process, and serine hydrolase activity, etc. (Fig. 4d). The relevant top-enriched up-KEGG pathways indicated that the Gel-co-MGFF enhanced cell adhesion and proliferation (Focal adhesion, PI3K-Akt signaling pathway) as well as matrix formation (ECM-receptor interaction, glycosaminoglycan biosynthesis-heparan sulfate/heparin) (Fig. 4e). A specific gene heatmap and its corresponding protein-protein interaction networks elucidated the relevant up- or down-regulated genes in GO and KEGG analysis, including upregulated Collagen II (*Col II*), and downregulated matrix metalloproteinase 13 (*Mmp-13*) (Fig. 4f, g). The transcriptomic analysis indicated that Gel-co-MGFF could potentially bind integrin family receptors to promote BMSCs adhesion, and then facilitate the ECM-receptor interaction, cell proliferation, and chondrogenic differentiation through focal adhesion, PI3K-Akt signaling, and hedgehog signaling pathway (Fig. 4h).

## Chondrogenic differentiation of BMSC-laden Gel-co-MGFF
To investigate Gel-co-MGFF-driven chondrogenic differentiation of BMSCs and confirm the results of the transcriptome analysis, we evaluated the expression of cartilage-related genes and the secretion of cartilage-relevant matrix. Immunofluorescent (IF) staining and

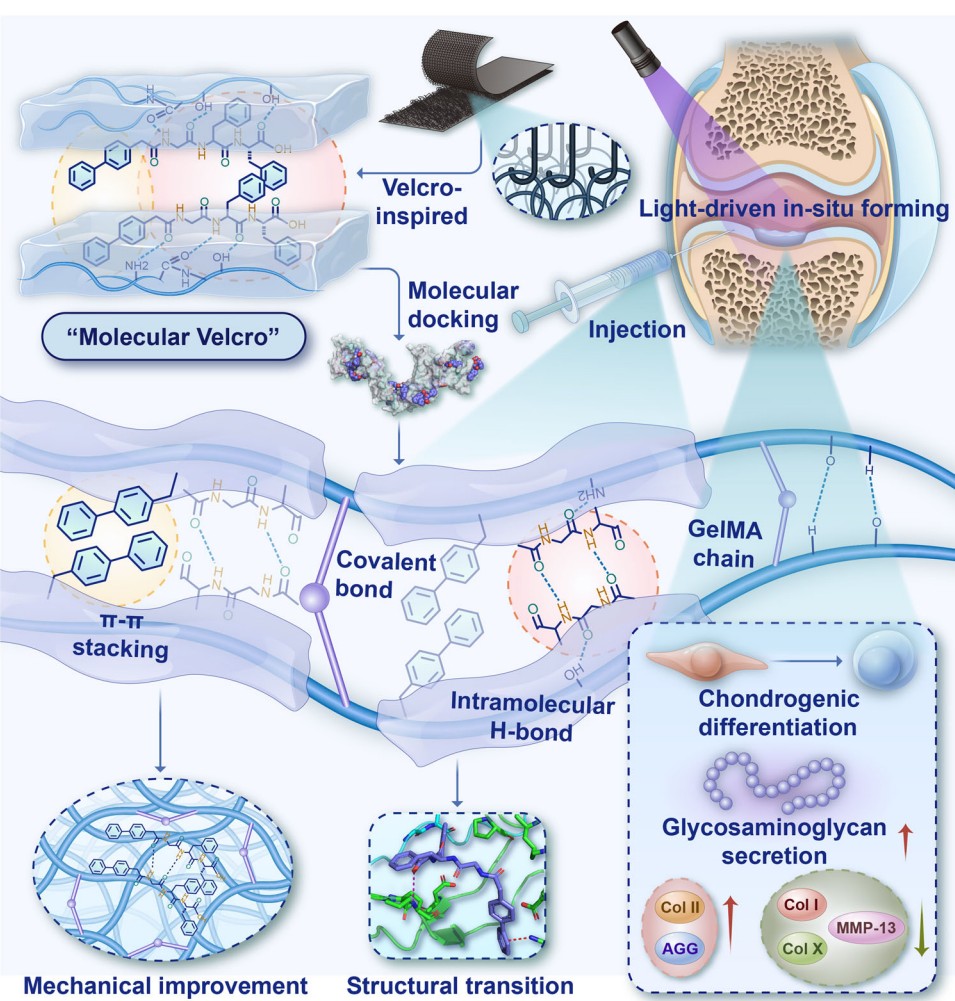

**Fig. 1 | Schematic diagram of velcro-inspired protein-peptides co-assembly strategy for constructing injectable fibrillar protein hydrogel enhancing in situ cartilage regeneration.** BPAA-GFF, as "molecular velcro", attached to UV-curable gelatin long chains via various intermolecular interactions as molecular docking indicated. Abundant intra- and intermolecular interaction (eg. Pi-Pi stacking and H-bond) drove the transition to a more ordered β-sheet conformation, inducing a more compact fibrillar structure of the co-assembled system, which substantially improved the mechanical strength of the protein gels (10-fold or more). This co-assembled hydrogel could achieve in-situ forming in the defective area of cartilage, and promote hyaline cartilage regeneration by promoting cell proliferation and matrix maintenance.

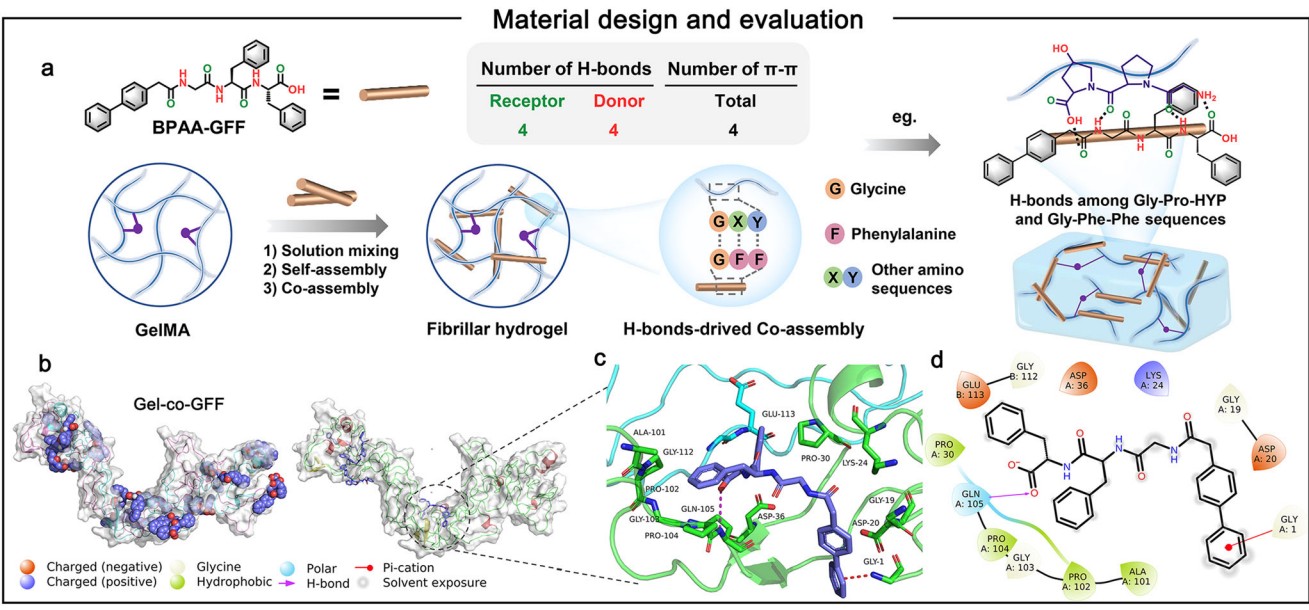

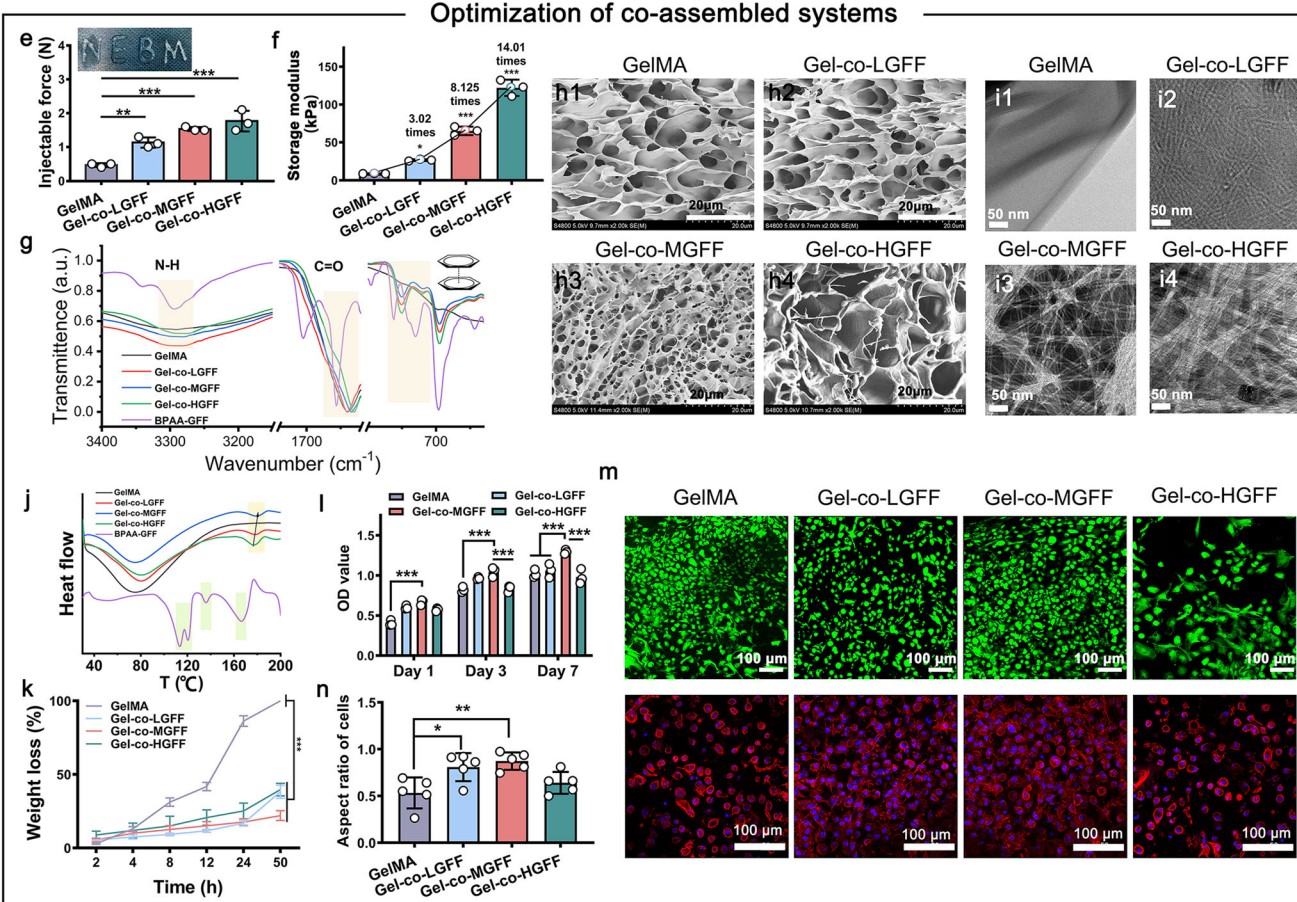

**Fig. 2 | Design, preparation, and characterization of co-assembled hydrogels. a** Schematic diagram of the co-assembly process between BPAA-GFF and GelMA. BPAA-GFF has four H-bonds donor or receptor and four Pi-Pi interaction site, respectively. It could achieve self-assembly and co-assembly with GelMA long chains by combining with abundant amino sequence of GelMA, eg. Gly-Pro-HYP. **b** Representative molecular docking images based on energy minimization for BPAA-GFF/GelMA assembly. **c, d** 3D and 2D pictures of representative interaction sites between BPAA-GFF and GelMA. **e** The injectable force of co-assembled hydrogels (**p = 0.007, ***p < 0.001) n = 3 independent samples. **f** The storage modulus of co-assembled hydrogels in 10 Hz (*p = 0.03, ***p < 0.001) n = 3 independent samples. **g** FT-IR spectra indicating H-bonding and Pi-Pi interactions.

**(h1-h4)** SEM images observing inner porous structure. n = 3 independent samples. **(i1-i4)** TEM images observing microscopic fibrillar morphology. n = 3 independent samples. **j** DSC curve illustrating elevated endothermic peak in co-assembled systems (Exotherm upward). **k** Degradation behavior of samples in vitro. n = 3 independent samples. **l** BMSCs proliferation measured by CCK-8 kit. n = 3 biologically independent experiments. **m** Live/Dead (green for live cells, red for dead cells) and cytoskeleton (red for phalloidine, blue for nucleus) staining. **n** cell aspect ratio quantification (*p = 0.02, **p = 0.005). n = 5 biological independent replicates. Data are mean ± s.d. (One-way analysis of variance (ANOVA), followed by Tukey's multiple comparison post hoc test, *p < 0.05, **p < 0.01 and ***p < 0.001).

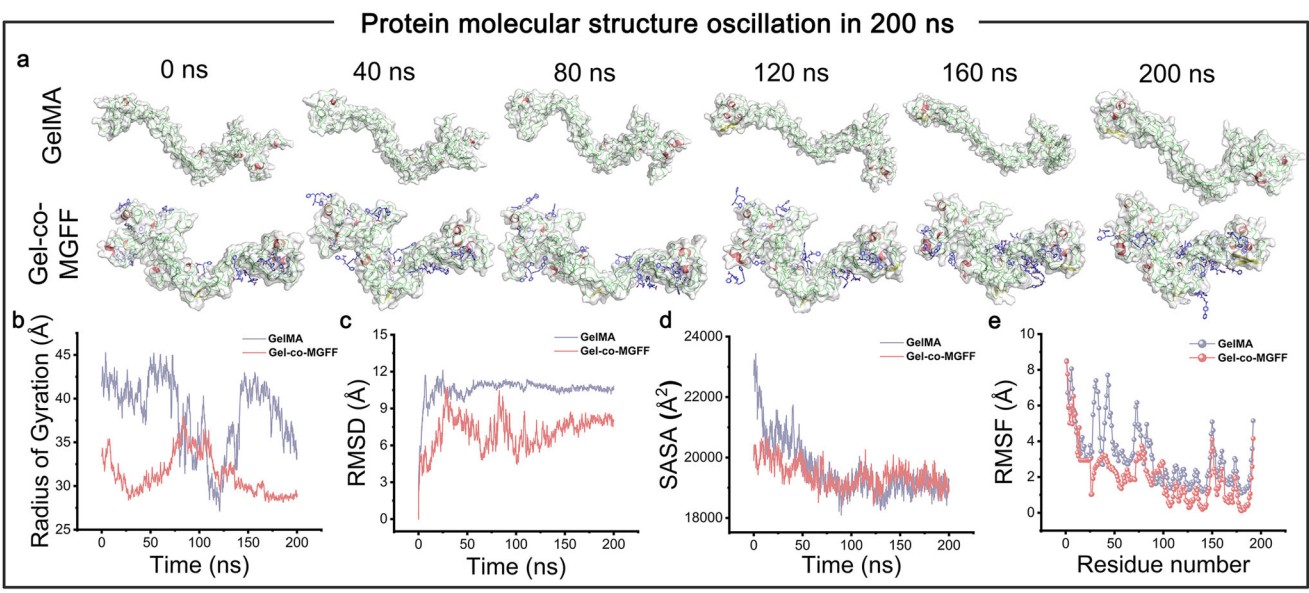

**Fig. 3 | Intermolecular interactions resulted in advanced structural transition of co-assembled hydrogels. a** Structure fluctuation during molecular dynamics simulation in 200 ns. **b–e** Fluctuation of Rg, RMSD, SASA, and RMSF during 200 ns simulation. **f1** and **f2** Interactions in GelMA and Gel-co-MGFF systems. **f3** Interactions between GelMA and BPAA-GFF. **g** Secondary structure content calculated from molecular simulation. **h** UV absorption spectrum indicating Pi-Pi stacking between BPAA-GFF and GelMA. (**i1–i3**) ThT fluorescent staining for β-sheet.

**j** ThT semi-quantification (**$p = 0.004$). $n = 3$ independent samples. **k1–l2** Relative content of secondary structure in GelMA and Gel-co-MGFF obtained from FT-IR spectra. **m** Schematic diagram of the structure and properties optimization during the co-assembly process between BPAA-GFF and GelMA. Statistical analyses were performed with one-way analysis of variance (ANOVA), followed by Tukey's multiple comparison post hoc test, *$p < 0.05$, **$p < 0.01$ and ***$p < 0.001$.

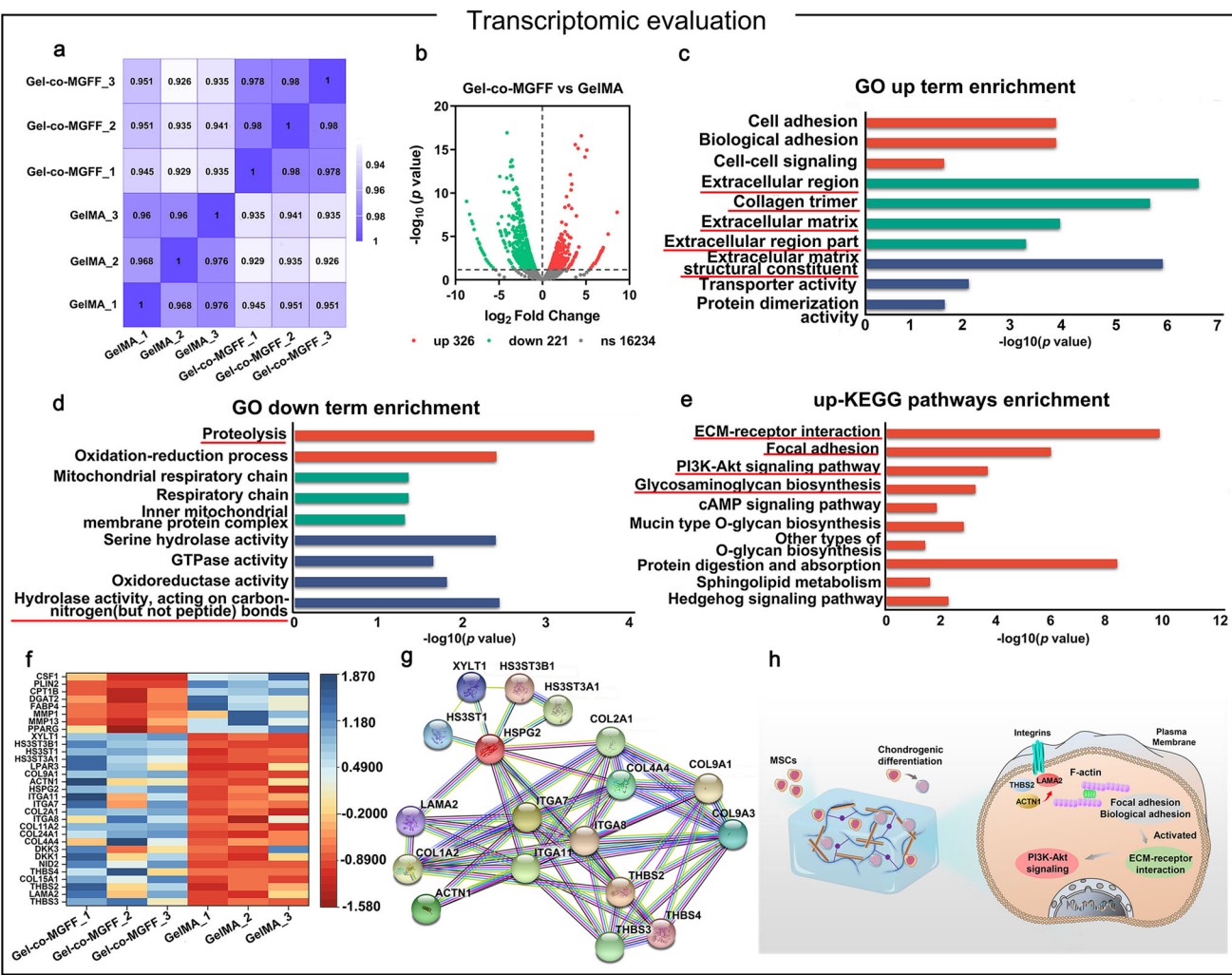

**Fig. 4 | Transcriptomic analysis revealing co-assembled hydrogel regulating BMSCs behavior. a** Heatmap of Pearson correlation. **b** Volcano plot of differentially expressed genes. **c**, **d** Up-regulated and down-regulated terms in enriched GO analysis of Gel-co-MGFF vs. GelMA analysis. **e** Up-regulated terms in enriched KEGG analysis of Gel-co-MGFF vs. GelMA analysis. **f** Heatmap analysis of differentially expressed genes involved in cell adhesion, ECM construction, and glycosaminoglycan synthesis of Gel-co-MGFF vs. GelMA analysis. **g** String interaction network for representative genes. **h** Schematic depiction of the potential mechanism of pathway regulation.

semi-quantitative analysis revealed that Gel-co-MGFF facilitated BMSCs to secrete more Collagen II (Col II) and SRY-box transcription factor 9 (Sox9), as well as less Collagen I (Col I) and Collagen X (Col X), suggesting that Gel-co-MGFF constructed a more suitable microenvironment for BMSC chondrogenic differentiation than GelMA (Figs. 5a, b1–b4). Furthermore, a larger amount of glycosaminoglycan (GAG) secretion and a smaller quantity of *Mmp-13* expression in Gel-co-MGFF verified its effect on hyaline cartilage matrix maintenance[24] (Fig. 5c, d). In addition, higher expression of Aggrecan (*Agg*) and *Col II* as well as lower expression of *Col I* and *Col X* revealed the ability of Gel-co-MGFF to promote chondrogenic differentiation of BMSCs and potentially restrain cartilage fibrosis and hypertrophy (Fig. 5e, f).

### Ectopic chondrogenesis and immune response in vivo

Ectopic chondrogenesis is often used as a key indicator to evaluate cartilage regeneration potential[25]. BMSC-laden Gel-co-MGFF or BMSC-laden GelMA was implanted subcutaneously into nude mice (Fig. 6a). After implantation for 30 days, a higher GAG to total DNA ratio indicated that Gel-co-MGFF promoted the formation and maintenance of the cartilage matrix (Fig. 6b). Quantitative polymerase chain reaction (q-PCR) and IF staining both demonstrated the effects of Gel-co-MGFF on hyaline cartilage phenotype maintenance (Fig. 6c–e,

Supplementary Fig. 8). Lower immunogenicity and an appropriate inflammatory response are known to accelerate tissue regeneration[26–28]. Therefore, it is necessary to evaluate the immunogenicity and inflammation induced by Gel-co-MGFF. IF staining and semi-quantitative analysis presented less mature B lymphocytes and immunoglobulin G (IgG) secretion in GelMA and Gel-co-MGFF (Figs. 6f and 5g), indicating that the two hydrogels were less immunogenic than animal-derived type I collagen. Furthermore, higher CD206+ expression (M2 macrophages) and lower CD86+ expression (M1 macrophages) indicated that Gel-co-MGFF produced a lower inflammatory response compared with GelMA (Fig. 6h, i).

### Cartilage reconstruction by BMSC-laden Gel-co-MGFF in a rabbit model

The effect of Gel-co-MGFF on articular cartilage regeneration was evaluated systematically in a full-thickness cartilage defect model (3 mm diameter) of rabbits. BMSC-laden hydrogels were injected into the articular cartilage defective area, followed by irradiation with blue light to crosslink hydrogels in situ (Supplementary Fig. 9a–c). After three months, the defect was almost entirely filled with regenerated cartilage in the Gel-co-MGFF group, and the International Cartilage Repair Society (ICRS) score of Gel-co-MGFF was higher than that of

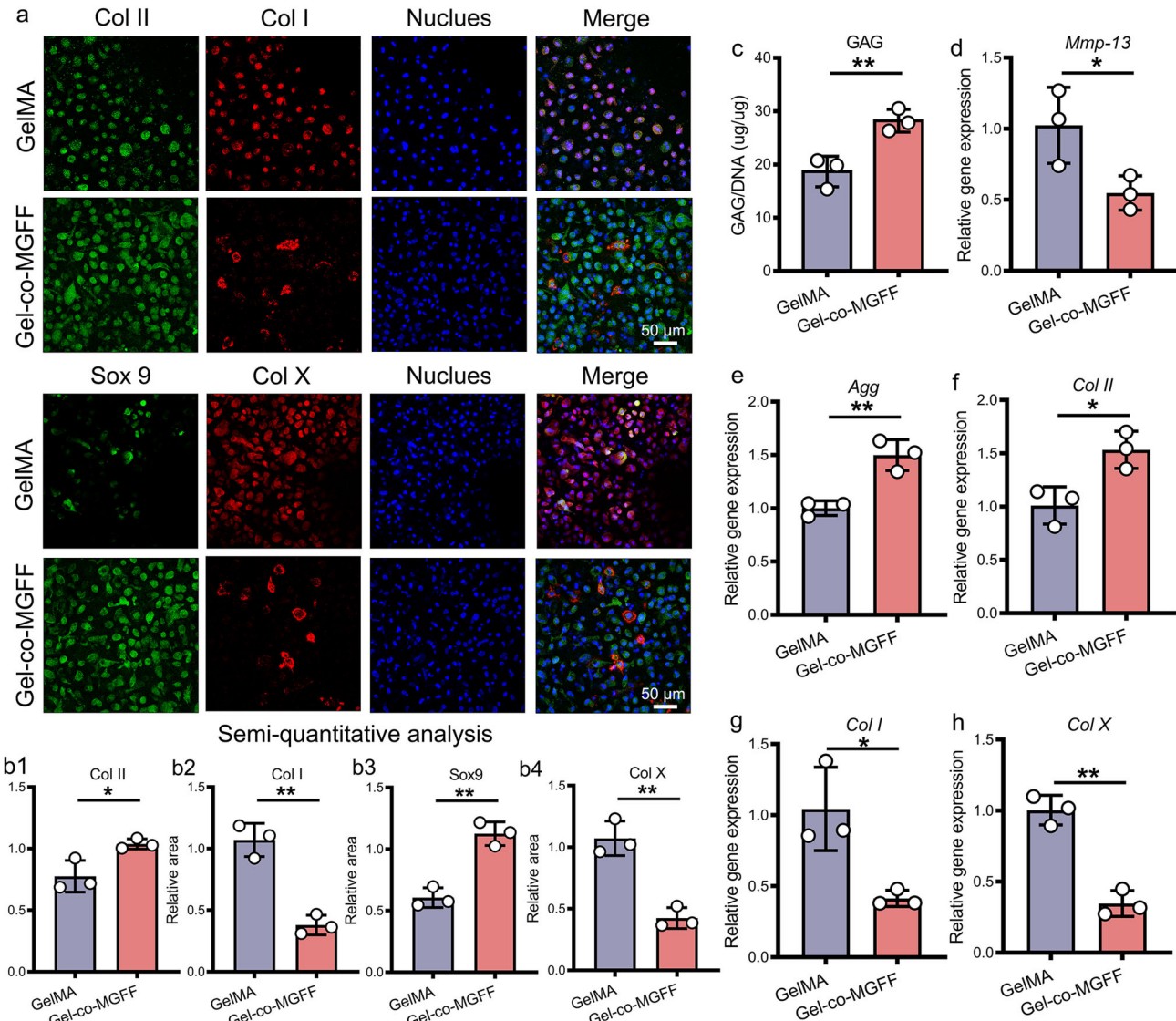

**Fig. 5 | Gel-co-MGFF promoted BMSC chondrogenic differentiation. a** CLSM images of Col I, Col II, Sox9, and Col X staining in vitro (scale bar: 50 μm) (*n* = 3 independent samples), and (**b1**–**b4**) semi-quantification (normalized to nucleus staining area) (**p* = 0.0282 ***p* = 0.0016 ***p* = 0.0020 ***p* = 0.0024). *n* = 3 independent samples. **c** Quantitative determination of GAG/DNA (***p* = 0.0098). *n* = 3 independent samples. **d**–**h** Chondrogenic gene expression of *Agg*, *Col II*, *Mmp-13*, *Col I*, and *Col X* of BMSCs cultured in co-assembled hydrogels for 14 days by RT-PCR (the expression level was normalized to *Gapdh*) (**p* = 0.0479, ***p* = 0.0058, **p* = 0.0215, **p* = 0.0217, ***p* = 0.0012) *n* = 3 independent samples. Data are mean ± s.d. Statistical analyses were performed with two-tailed unpaired t tests, **p* < 0.05, ***p* < 0.01 and ****p* < 0.001.

GelMA (Fig. 7a–c). The cartilage thickness and collagen fiber direction in the Gel-co-MGFF group were similar to the normal group (Fig. 7d–f, Supplementary Fig. 9d). Masson's trichrome (MT) staining and Safranin O-Fast green (SO-FG) staining revealed more collagen and proteoglycan secretion in Gel-co-MGFF (Fig. 7g–i, Supplementary Fig. 9e, f). However, there was an obvious defect in blank group after three months. The disordered arrangement of cells and collagen fiber as well as limited ECM secretion also confirmed the ineffective cartilage regeneration of blank group (Supplementary Fig. 9d–f), which was consistent with these literatures reported[29–31]. Compared to GelMA, more secretion of proteoglycan 4 (Prg4) illustrated the effect of Gel-co-MGFF on reducing cartilage friction (Fig. 7j, k). Higher expression of Col II, as well as less secretion of Col I and Col X, confirmed the ability of BMSCs-laden Gel-co-MGFF to promote hyaline cartilage regeneration and restrain cartilage fibrosis and hypertrophy in vivo. Furthermore, less secretion of Mmp-13 confirmed the improved biofunction of Gel-co-MGFF in cartilage matrix maintenance (Fig. 7l–s, Supplementary Fig. 10).

## Discussion

Stem cell-laden injectable hydrogels can provide an effective solution for in situ cartilage regeneration under minimally invasive surgery[32,33]. Due to their extracellular matrix-like properties, hydrogels can provide suitable niches for stem cells, thus regulating the influence of stem cells on cartilage regeneration[34–36]. As one of the extracellular matrix components, collagen has been widely used as a scaffold in cartilage engineering due to its unique fibrous structure and has shown good biological performance. However, the clinical applications of collagen are limited by potential immunogenicity and rapid degradation behavior, making it unsuitable as a long-term stable microenvironment for stem cells. Collagen derivatives with low immunogenicity (e.g., inactivated collagen, humanized recombinant collagen, and collagen mimetic peptides) have been extensively investigated to mimic the properties of collagen[4,37–39]. However, the mechanical strength and stability of these derivatives were insufficient due to the random arrangement of amino acid molecular chains and the lack of sufficient cross-linking sites. Research showed that the mechanical strength and

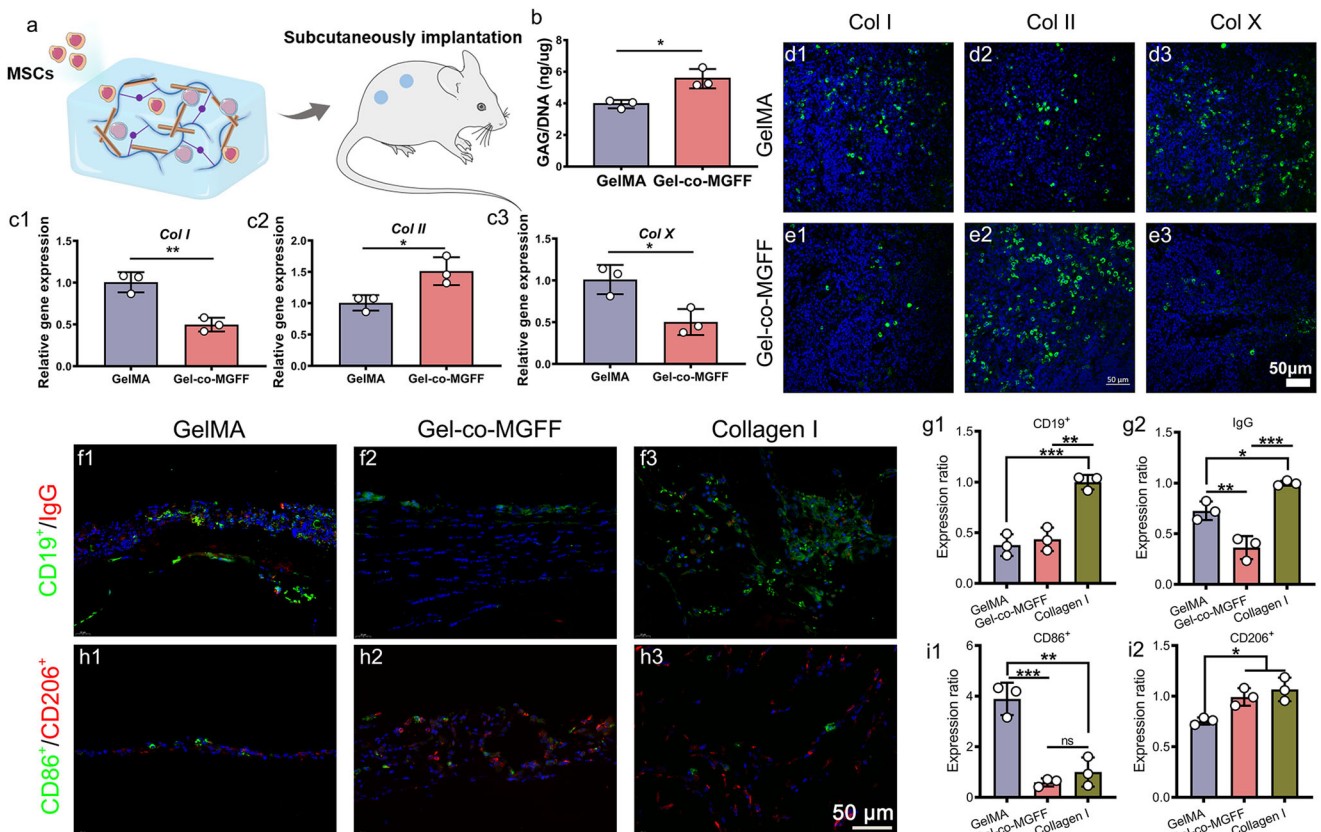

**Fig. 6 | Ectopic chondrogenesis and immune response in vivo. a** Schematic diagram of BMSCs-laden co-assembled hydrogels transplanted into nude mice subcutaneously. **b** Quantification of GAG/DNA after 30 days of implantation (*$p$ = 0.01) $n$ = 3 independent samples. **c** Chondrogenic gene expression of *Col I*, *Col II*, and *Col X* after 30 days of implantation (the expression level was normalized to *Gapdh*) (**$p$ = 0.004, *$p$ = 0.03, *$p$ = 0.02) $n$ = 3 independent samples. (**d1–e3**) CLSM images for Col I, Col II, and Col X staining. $n$ = 3 biologically independent samples. **f1–f3** Immunostainings of IgG secretion and CD19[+] B lymphocytes around hydrogels after 7 days Intramuscular implantation in BALB/C mice, and (**g1** and **g2**) semi-quantification (normalized to collagen I group) (**$p$ = 0.001, **$p$ = 0.005, *$p$ = 0.02, ***$p$ < 0.001) $n$ = 3 biologically independent samples. (**h1–h3**) Immunostainings of CD86[+] M1 macrophages and CD206[+] M2 macrophages, and (**i1** and **i2**) semi-quantification (normalized to collagen I group) (**$p$ = 0.001, ***$p$ < 0.001, *$p$ = 0.03, *$p$ = 0.01). $n$ = 3 biologically independent samples. Data are mean ± s.d. Statistical analyses between two groups were performed with Student's unpaired t tests and between three or more groups with one-way analysis of variance (ANOVA), followed by Tukey's multiple comparison post hoc test, *$p$ < 0.05, **$p$ < 0.01 and ***$p$ < 0.001.

stability of protein gels could be significantly improved by regulating the hydrogen bonds[40], electrostatic interactions[4], and hydrophobic interactions[4] between molecular chains. In addition, intermolecular interactions such as hydrogen bonding between amino acids have been demonstrated to provide the primary driving force for conformational transitions of proteins or peptides[41,42]. Studies reported that a higher content of β-sheet could be induced by altering the interactions between amino acid chains[43]. This locally ordered, spatial conformation contributed to improving the mechanical strength of hydrogels while endowing them with good cytocompatibility[44].

Here, inspired by velcro, we have introduced BPAA-GFF, a functional short peptide that promotes chondrocyte proliferation and phenotype maintenance, into UV-curable gelatin to construct a co-assembled system through intermolecular interactions such as hydrogen bonding (Fig. 2a). This strategy of optimizing the assembly behavior of the system through GelMA-short peptide interactions enhanced the β-sheet content (Fig. 3f–l), which, in turn, substantially improved the compression modulus and stability of the system (Fig. 2f, j). Compared with other strategies, this system was based only on the regulation of intermolecular interactions between short peptides and gelatin chains, free of additional ions and complex post-processing, which greatly improved the biosafety and activity of the protein.

First, using the techniques of molecular docking, we confirmed that the "molecule velcro" BPAA-GFF attached to GelMA via various H-bonds, Pi-cation, salt bridge, and Pi-Pi stacking interactions, which demonstrated the feasibility of our strategy (Fig. 2a–d). These intermolecular interactions led directly to a significant increase in the compression modulus 10-fold or more (Fig. 2f). The co-assembled system was further characterized by analyzing its internal pores, fibrillar structure, and effects on cell proliferation and phenotype maintenance (Fig. 2h–n). We demonstrated that Gel-co-MGFF possessed a more regular internal pore structure and microscopic fibrillar network compared with Gel-co-LGFF and Gel-co-HGFF, and internal BPAA-GFF molecules were combined with GelMA in a more stable manner (Fig. 2j). Literature reported that the small pore size (60–125 μm) was an important factor in BMSCs aggregation and subsequent chondrogenic differentiation. However, the larger pore size (425–600 μm) was critical for vascularization and subsequent bone formation[45,46]. As shown in SEM images (Fig. 2h), small pore sizes (around 20 μm) were exhibited in four groups, which is close to the reported small pore size (60–125 μm) for facilitating cartilage regeneration[46]. This might be one of the potential reasons for the maintenance of chondrocyte phenotype in vitro and cartilage regeneration in vivo. Furthermore, molecular dynamics simulations were applied to investigate the kinetic characteristics of intermolecular interactions within a certain time period. The "molecular velcro" effect, driven by hydrogen bonding and hydrophobic interactions, led to the formation of more ordered β-sheet regions in the system, resulting in a more compact structure and enhanced stability. Such ordered spatial

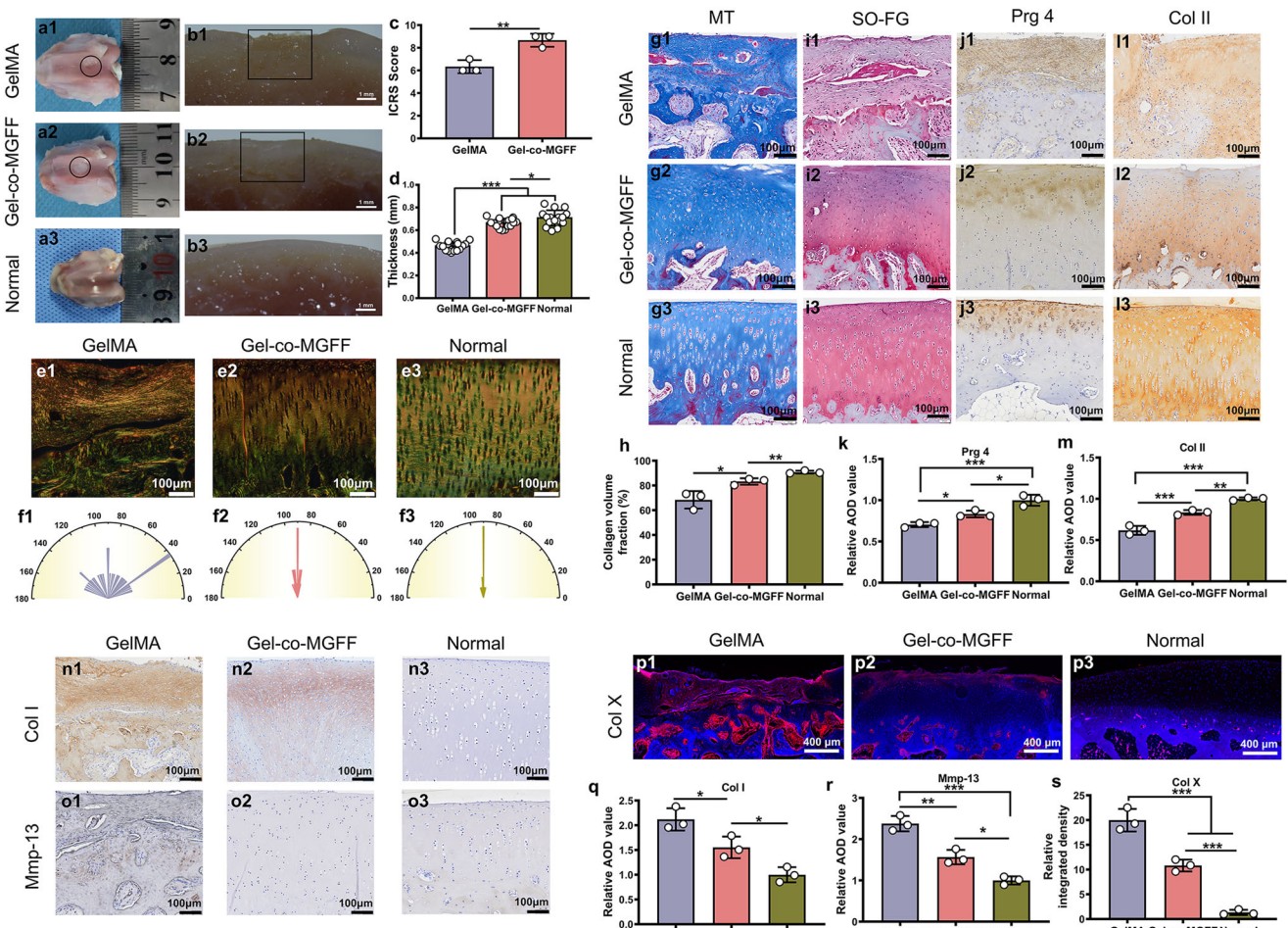

**Fig. 7 | BMSCs-laden Gel-co-MGFF regenerated hyaline cartilage in a rabbit model. a1–a3** Representative images of the defective area after 3 months of implantation. **b1–b3** Representative pictures for the sagittal plane of the knee-joints defective area. **c** ICRS score evaluating the effect of cartilage repair comprehensively (**$p = 0.008$), $n = 3$ independent samples. **d** Cartilage thickness statistics (*$p = 0.05$, ***$p < 0.001$) $n = 16$ independent tests. **e1–e3** Sirius-red staining ($n = 3$ biologically independent replicates), and **f1–f3** collagen fiber orientation statistics. **g1–g3** MT staining and **h** semi-quantification (*$p = 0.01$, **$p = 0.002$). $n = 3$ biologically independent replicates. **i1–i3** SO-FG staining ($n = 3$ biologically independent replicates). **j1–j3** Immunohistochemistry staining for Prg 4 and (**k**) semi-quantification (*$p = 0.04$, *$p = 0.01$, ***$p < 0.001$) (normalized to normal group),

$n = 3$ biologically independent replicates. **l1–l3** Immunohistochemistry staining for Col II and (**m**) semi-quantification (***$p < 0.001$, ***$p < 0.001$, **$p = 0.004$) (normalized to normal group, $n = 3$ biologically independent replicates). **n1–n3** Immunostaining for Col I. **o1–o3** Immunostaining for Mmp-13. **p1–p3** Immunostaining for Col X. **q–s** semi-quantification (*$p = 0.03$, *$p = 0.04$, *$p = 0.01$, **$p = 0.002$, ***$p < 0.001$) (normalized to normal group, $n = 3$ biologically independent replicates). Data are mean ± s.d. Statistical analyses between two groups were performed with Student's unpaired t tests and between three or more groups with one-way analysis of variance (ANOVA), followed by Tukey's multiple comparison post hoc test, *$p < 0.05$, **$p < 0.01$ and ***$p < 0.001$.

conformational transitions might potentially affect biological functions.

Transcriptomic analysis revealed that the co-assembled system promoted cell proliferation and matrix secretion through the upregulation of biological pathways including the PI3K-Akt signaling pathway, ECM-receptor interaction, and glycosaminoglycan biosynthesis (Fig. 4c–e) while promoting chondrogenic differentiation and matrix maintenance[24]. The higher expression of *Col II* and GAG as well as the lower expression of *Mmp-13* in Gel-co-MGFF validated the biological effect revealed by the transcriptome analysis. Furthermore, the lower secretion of Col I and Col X implied better phenotype maintenance of hyaline cartilage (Fig. 5).

Next, we showed that Gel-co-MGFF promoted ectopic chondrogenesis of BMSCs and hyaline cartilage phenotype maintenance after subcutaneous implantation of BMSCs-laden hydrogel into nude mice (Fig. 6a–e). Less B lymphocytes activation and IgG secretion demonstrated the lower immunogenicity of the co-assembled system (Fig. 6f, g). Fewer M1 macrophages and more M2 macrophages illustrated the anti-inflammatory properties of the Gel-co-MGFF, which

might be because BPAA itself was an anti-inflammatory agent (Fig. 6h, i). After three months of implantation in the rabbit cartilage defective area, the thickness (Fig. 7d) and collagen fiber arrangement (Fig. 7e, f) of regenerated cartilage in Gel-co-MGFF were more similar to that of natural cartilage. Compared with GelMA, Gel-co-MGFF could better maintain the hyaline cartilage phenotype in vivo, inhibiting hypertrophy and fibrosis of cartilage. More lubricating protein Prg4 was secreted in Gel-co-MGFF, potentially reducing the friction of articular cartilage (Fig. 7j, k). Meanwhile, Gel-co-MGFF could also promote cartilage matrix formation and inhibit degradation, further supporting the biological mechanism revealed by the transcriptome analysis. (Fig. 7).

In summary, as a "molecular velcro", BPAA-GFF molecules reconstructed the spatial conformation of UV-curable gelatin long chains, thus forming a stable co-assembled protein hydrogel with more advanced biofunctions. Supramolecular short peptide fibers drove the transition to a more ordered β-sheet conformation, inducing a more compact fibrillar structure of the co-assembled system, which substantially improved the mechanical strength of the protein

gels (10-fold or more). Both in vitro and in vivo biological evaluations confirmed that Gel-co-MGFF promoted cartilage regeneration by promoting cell proliferation and matrix maintenance. This "molecular velcro" concept provides a brand-new strategy for the development of injectable protein gels with high strength for cartilage regeneration.

## Methods

### Materials

Gelatin was purchased from Avantor, Inc. Methacrylic anhydride (MA) was purchased from Aladdin Biochemical Technology Co., Ltd. (Shanghai, China). N, N-Diisopropylethylamine (DIPEA), 4-biphenyl acetic acid (BPAA), and trifluoroacetic acid (TFA) were purchased from Best-reagent Corporation (Chengdu, China). Fmoc-L-Phe-OH, Fmoc-Gly-Oh, and 2-chlorotrityl chloride resin were provided from Shanghai GL biochem (Shanghai, China). Phosphate buffered saline (PBS) was bought from Invitrogen. Dulbecco's Modified Eagle Medium (DMEM), fetal bovine serum (FBS), type II collagenase, trypsin, and penicillin/streptomycin (PS) were bought from Gibco (USA). α-MEM medium was purchased from Hyclone (USA). Type I collagen was extracted and purified from new-born calf skin in our laboratory. All other chemicals were used as received unless otherwise specified.

### Preparation of GelMA

First, 1 g gelatin was dissolved in 100 mL PBS at 50 °C. Then, 1 mL methacrylic anhydride solution was added followed by adjusting the pH of the solution to 8.5 ~ 9 with 1 M NaOH. After 3 h, the reaction was finished by adjusting the solution pH to 7.4. Then the product was centrifuged at 722.75 ×g for 5 min to remove residual methacrylic anhydride. Next, the solution was dialyzed (Mw cutoff 8000–14000) against deionized water for 3 days. Finally, the GelMA product was obtained after lyophilization. The chemical structure of GelMA were analyzed by $^1$H NMR (400 MHz, Bruker AMX-400, USA) and MestRe-Nova Version 10.0.1 software.

### Synthesis of BPAA-GFF

BPAA-GFF was synthesized following solid phase peptide synthesis (SPPS)[14,15]. In brief, a dichloromethane (DCM) solution dissolving Fmoc-L-Phe-OH and DIPEA was introduced into 2-chlorotrityl chloride resin and mixed. After the coupling process, the resin was washed with DCM and quenched by using a mixed solution (DCM/methanol/DIPEA = 7/2/1, $v/v$) for 10 min. Then, the Fmoc-protecting group was removed by 20% piperidine solution in N, N-Dimethylformamide (DMF, $v/v$) for 30 min. Next, the subsequent another Fmoc-L-Phe-OH in DMF was introduced with the assistance of 1.5 mmol O-(Benzotriazol-1-yl)-N, N, N', N'-tetramethyluronium Hexafluorophosphate (HBTU). The process of coupling Fmoc-Gly-OH and deprotecting Fmoc group was repeated as mentioned above. Subsequently, BPAA was connected to the peptide chain by using 1.5 mmol HBTU and 3.0 mmol DIPEA dissolved in DMF. Finally, the coarse product was separated from the resin by applying DCM/TFA (99:1, $v/v$) and purified by precipitation in cold diethyl ether. The chemical structure was identified by mass spectroscopy (MS).

### Preparation of GelMA/BPAA-GFF co-assembled hydrogels

First, BPAA-GFF was dissolved in ultrapure water at a concentration of 25 mM, 50 mM, and 75 mM, respectively. Then, the solution pH was adjusted to 10–11 by adding 1 M NaOH to make sure complete dissolution. The pH value was adjusted to 7–8 when BPAA-GFF was completely dissolved. Next, the GelMA (5.0%, $w/v$) and LAP (0.5%, $w/v$) were introduced into the solution and then the solution was deposited at 37 °C for 20 min. Finally, the hydrogels were formed by irradiating with blue light at the wavenumber of 405 nm.

### Characterization of GelMA/BPAA-GFF co-assembled hydrogels

The morphology of the hydrogels was observed by SEM (HITACHI S-800, Japan). The inner nanostructures of hydrogels were observed by TEM (TECNAI G2 F20 S-TWIN at 200 kV). The chemical structure of co-assembled hydrogels was characterized by FT-IR spectroscopy (Nicolet 6700, USA) and DSC (METTLER TOLEDO, Switzerland) with air flow (25 ~ 200 °C, 10 °C/min). The injection force was measured using an electromechanical universal testing machine with a maximum load of 500 N (Shimadzu Autograph AGS-X, Japan) at 30 mm/min crosshead displacement. 1 mL syringes filled with hydrogels were applied to measure the injection force. Three samples were employed to measure.

The disintegration performance was measured under simulated physiological conditions. First, four groups of hydrogels were prepared as the method above. Then, the initial weight of the samples was recorded ($W_o$). Next, the samples were immersed in the solution containing Type II collagenase (100 units/mL). After that, they were placed in a constant temperature shaker at 40 rpm at 37 °C. The hydrogels were taken out and the excess solution on the surface was removed by filter paper at a certain interval. Then the weight ($W_b$) was determined. The disintegration of the hydrogels was calculated following the formula: Weight loss percentage = $(W_o − W_b)/W_o$. The storage modulus (G′) and loss modulus (G″) were measured by a dynamic mechanical analyzer (DMA, TA-Q800, USA) varying the frequency (a frequency shift from 1 Hz to 10 Hz, an amplitude of 40 μm, a preload force of 0.002 N and a force track of 105%) at 25 °C. Three samples were measured in three tests. The ultraviolet-visible (UV) absorption spectra was acquired on a UV spectrometer (Lambda 850, PerkinElmer, USA). All the data were collected with a bandwidth of 1 nm, a step of 0.01 nm and the fluctuation of wavelengths between 0 and 700 nm. The ThT (Maclin, China) staining was achieved as follows. Hydrogels were immersed in 5 mM ThT solution for 1 h, followed by being observed with a confocal laser scanning microscopy (CLSM) (LSM 880, Zeiss, Germany).

### Molecular docking and molecular dynamics simulation

The amino sequence of gelatin was screened from the UniProt database according to the pig collagen type II alpha 1 chain. The specific amino sequence (GPKGPPGPQGPAGEQGPRGDRGDKGEK GAPGPRG RDGEPGTPGNPGPPGPPGPPGPPGLGGNFAAQMAGGFDEKAGGAQMGV MQGPMGPMGPRGPPGPAGAPGPQGFQGNPGEPGEPGVSGPMGPRGPP GPPGKPGDDGEAGKPGKSGERGPPGPQGARGFPGTPGLPGVKGHRGYPG LDGAKGEAGAPGVK, total 192 amino acids) was adopted to conduct the simulation process. For molecular docking, water molecules and irrelevant heteroatoms were removed using UCSF Chimera to preserve the protein structure alone. Protein atomic charges were calculated using AMBER14SB, and amino acid PK values were calculated and assigned under neutral conditions using H++3 online tool. The modified peptide structures were generated in 3D by the open-source cheminformatics package RDKit. And the conformation sampling was performed. The conformation was optimized using MMFF94 force field and the low-energy conformation was finally output. AM1-BCC local charges were assigned using UCSF Chimera. The best binding site was predicted using SiteMap software.

For molecular dynamics simulation, 3D protein structures were constructed using Alphafold2. The protein dimer cross-linked structure was constructed using Avogadro software. Methacrylic anhydride crosslinking modifications were performed according to the lysines exposed on the protein surface. In this study, molecular dynamics simulations were performed using the Gromacs 5.1.5 open-source software package. The simulated system was set in a confined environment with the temperature set to 300 K, pH set to 7.4, and the pressure to one atmosphere 1 bar. TIP3P water molecule was adopted, and the system charge was balanced using NaCl solvent. After the initial system construction, the steepest descent method was applied

to all atoms to minimize the system energy. The system is subjected to kinetic simulations of 200 ns duration and simulated every 2 fs. After all simulations were completed, the papameters of Rg, SASA, RMSD, and RMSF were calculated.

## Construction of 3D BMSC-laden hydrogels

BMSCs were extracted from the new-born rabbit. The bilateral femur, tibia, and humerus were removed under sterile conditions after euthanasia and disinfection. Then, alpha-modified Eagle's medium (α-MEM, 22571038, Gibco) was injected into the bone marrow cavity to flush out the BMSCs after the epiphysis was removed with scissors. BMSCs were cultured with α-MEM medium in which containing 20% fetal bovine serum (FBS, 10100147, Gibco) and 1% penicillin/streptomycin (10378016, Gibco). The BMSCs were washed with PBS until the BMSCs adhesion. Then, the BMSCs were expanded to passage three for use. Precursor solutions were prepared as method mentioned above with germfree BPAA-GFF and GelMA. Normally, 0.05% trypsin/EDTA was applied to trypsinize and resuspend BMSCs. Then, BMSCs were suspended into the precursor solutions with a final concentration of $5 \times 10^6$ cells per mL. Four groups mixtures were injected into silastic molds where they were irradiated with blue light for crosslink, and then incubated at 37 °C for half an hour. Then the hybrid hydrogel/cell complexes were cultured in 24-well plate with 1.5 mL chondrogenic differentiation medium under same conditions (5% $CO_2$, 37 °C). The medium was replaced every 2 days.

## RNA sequencing

BMSCs were encapsulated into hydrogels as mentioned above and cocultured for 14 days in vitro. Transcriptome analysis for the specimens in vitro was performed by Novogene. After RNA extraction and detection of RNA degradation and contamination, a total amount of 1 μg RNA per sample was used as input material for the RNA sample preparations. Sequencing libraries were generated using NEBNext UltraTM RNA Library Prep Kit for Illumina (NEB, USA). The clustering of the index-coded samples was performed on a cBot Cluster Generation System using TruSeq PE Cluster Kit v3-cBot-HS (Illumia) according to the manufacturer's instructions. After cluster generation, the library preparations were sequenced on an Illumina Novaseq platform and 150 bp paired-end reads were generated. Raw data (raw reads) of fastq format were first processed through in-house Perl scripts. In this step, clean data (clean reads) were obtained by removing reads containing adapter, and the downstream analyses were based on clean data. FPKM method was adopted to determine gene expression. DESeq algorithm was applied to calculate the differentially expressed genes. Significant analysis was performed using the *p*-value and false discovery rate (FDR) analysis. Meanwhile, differentially expressed genes were determined as fold change > 2 or fold change <0.5, FDR < 0.05.

## Proliferation and morphology of BMSCs in hydrogels

Live/Dead staining and CCK-8 were applied to evaluate the proliferation of BMSCs cultured in hydrogels. For CCK-8 test, every hydrogel piece was incubated in 200 μL medium containing 10% CCK-8 (C0037, Beyotime, China) for 3 h, and OD value was recorded at 450 nm. As for the Live/Dead cell staining, the samples were soaked in PBS solution, which contained fluorescein diacetate (FDA, F1303, Invitrogen) (1‰, *v/v*) and propidium iodide (PI, P1304MP, Invitrogen) (1‰, *v/v*), and observed by CLSM. The cytoskeleton of the BMSCs in hydrogels was exhibited using rhodamine-phalloidin (1 h, 94072, Sigma-Aldrich) and DAPI (1 min, 28718-90-3, Sigma-Aldrich), which was imaged by CLSM.

## Influence of photocuring on BMSCs in vitro

BMSCs were cultured in 24-pore plate ($2 \times 10^4$ cells/pore) and divided into four groups, including BMSCs (without treatment), BMSCs + light (UV irradiation), BMSCs + LAP (0.5% LAP in medium, *w/v*), and BMSCs + LAP + Light (0.5% LAP in medium and initiated by UV light, *w/v*). After culture for 48 h, live/dead staining and CCK-8 test were applied to evaluate the cell viability and proliferation, respectively.

## Influence of hydrogels on chondrogenic differentiation of BMSCs in vitro

The GAG quantitative determination was conducted as follows. In short, the hydrogel/cell complexes were transferred into ep tubes after being cultured for 14 days. Then, they were washed by PBS. Subsequently, all hydrogel/cell complexes were immersed in phosphate buffer (200 mM, pH = 6.5) with papain. The samples were placed in constant temperature water bath at 65 °C overnight. Then, the supernatant by centrifugation (5900 ×g, 10 min) was collected for the measurement of GAG content by the Blyscan GAG assay kit (B100, Biocolor, U.K.) and the DNA content by Hoechst 33258 (B1302, Sigma-aldrich, China). The obtained data was calculated by the formula of standard curve fitting and presented in the form of the ratio of GAG to DNA.

RNA expression analysis was processed in the following way. The hybrid hydrogel/BMSCs complexes were collected in ep tubes containing RNAlater solution (Ambin) and stored at −20 °C. The RNA of samples was extracted with RNeasy Mini Kit (74104, Qiagen), and the concentration of RNA was detected by a microspectrophotomer (ND1000, Nanodrop Technologies). Then, iScript™ cDNA Synthesis Kit (BIO-RAD) was used to reverse transcript (RT) extracted RNA to cDNA, and SsoFast™ EvaGreen Supermix (BIO-RAD) was used to conduct polymerase chain reaction (PCR) that performed on a CFX96 Touch™ Real-Time PCR Detection System (BIO-RAD). The expression levels of *Col I*, *Col II*, *Col X*, *Agg* and *Mmp-13* were evaluated, and the primers were listed in Table S2. All the expressions of other genes were calculated on the basis of the expression level of *Gapdh*.

To further estimate the chondrogenic differentiation of BMSCs cultured in hydrogels, IF staining was carried out. Cultured for 14 days, the samples were washed with PBS and fixed by 4% paraformaldehyde (30525-89-4, CHRON CHEMICALS, China) for 3 days subsequently. After being treated with 0.2% Triton X-100 for 30 min and blocked with 5% goat serum at room temperature for 1 h, the samples were allowed to incubate overnight at 4 °C with the primary antibodies Col I (NOVUS, NB600-408, 1:200), Col II (NOVUS, NB600-844, 1:500), Col X (Invitrogen, MA5-14268, 1:200), and Sox9 (Bioss, bs-10725R, 1:200) as well as secondary antibodies Goat Anti-Mouse IgG (H + L) (Servicebio, GB21301, 1:200), Goat Anti-Rabbit IgG (H + L) (Servicebio, GB21303, 1:200), Goat Anti-Rabbit IgG (H + L) (Servicebio, GB22303, 1:200), and Goat Anti-Mouse IgG (H + L) (Servicebio, GB22301, 1:200) for 1 h by co-staining with DAPI for nuclei. After staining, CLSM was applied to observe the image.

## The in vivo immune response in mouse intramuscular implantation model

All animal studies were approved by the Sichuan University Medical Ethics Committee (approval number: KS2020330). All animal procedures were performed in accordance with the guidelines for care and use of Laboratory Animals of Sichuan University (approval number: KS2020330). Five male BALB/C mice (6 weeks, ~20 g) were sacrificed in this test. After mice were anesthetized by intraperitoneal injection of 3% pentobarbital sodium (30 mg/kg), hydrogels (Φ = 8 mm) were implanted into the bilateral thigh muscles, respectively. The implants (n = 3) were harvested 7 days post-surgery to analyze immune responses. IF staining was performed to detect CD19 (NOVUS, NBP2-15782, 1:500), IgG (Bioss, bs-0296R-Cy7, 1:500), CD86 (Affinity, DF6332, 1:400), and CD206 (Affinity, DF4149, 1:400) expression by co-staining with DAPI for nuclei. The sections were scanned by an automatic digital slide scanner and analyzed by the Case Viewer 2.1 software (Pannoramic MIDI, 3D HISTECH, Hungary). The

semi-quantitative results of fluorescence intensity were determined by using Image J software.

## The in vivo evaluation of chondrogenic differentiation and cartilage reconstruction

To test the expression of cartilage–related genes, 3D BMSC-laden hydrogels were transplanted subcutaneously in nude mice. In brief, after nude mice were anesthetized by intraperitoneal injection of 1% pentobarbital sodium (30 mg/kg), a minimally invasive incision on the skin was constructed on the back of nude mice, and subdermal areas were dissected with forceps. Then the cell-laden hydrogels were inserted into the subdermal area with no contact with each other. The incision was sutured and the mice were raised in SPF surrounding. After one month, the implants were taken out for GAG quantitative analysis, RT-PCR, and IF staining.

Seven adult New Zealand white rabbits (2.5–3.0 kg, male) were adopted and randomly divided into four groups (Blank, GelMA, Gel-co-MGFF, and Normal) for in vivo evaluation of cartilage reconstruction in this research. After being anesthetized by intravenous injection of 3% pentobarbital sodium (30 mg/kg), the experimental rabbits were fixed on the operating table in a supine position. The legs around the knee joints were shaved, disinfected, and draped. After an incision (about 1.5 cm long) was made on the inner side of the knee joint, the patella was everted to expose the trochlear groove. Then, a sterile hollow metal tube was applied to drill a circle (3 mm in diameter) on the center of the trochlear groove, and the hyaline cartilage was scraped out with a scalpel until the calcified cartilage layer was exposed. The animals were included in the study if they underwent successful hyaline cartilage removal, while the animals would be excluded if the subchondral bone was damaged and/or there was blood leaking during this process. For GelMA and Gel-co-MGFF groups, precursor solutions were injected into the defective area on the trochlear groove of each side and were cross-linked with initiating by LAP under the irradiation of blue light ($\lambda = 405$ nm). The defective area of rabbits in blank groups was treated without filler. All surgically treated rabbits were sutured and disinfected with iodine after patella repositioning. Surgical operation was not applied on the normal group rabbits. Each group has three duplicate samples. After euthanasia, all samples were taken out after 3 months, then the knee appearance was photographed. The macroscopic and histological results were analyzed by three investigators who were blind to the groups. Five different researchers were randomly involved in animal group distribution, operation, and evaluation. For histological analysis, samples were fixed in 4% paraformaldehyde and then soaked in 10% EDTA decalcifying solution (6381-92-6, CHRON CHEMICALS, China), which was changed every two days. Then, samples were paraffin sectioned at a thickness of 5 µm followed by MT (G1343, Solarbio), SO-FG (G1371, Solarbio) and Sirius red (PH1098, Scientific PHYGENE) staining. IHC and IF staining were further performed to analyze the chondrogenesis-related matrix, including Col I (NOVUS, NB600-408, 1:200), Col II (NOVUS, NB600-844, 1:500), Col X (Invitrogen, MA5-14268, 1:200), Prg4 (Sigma-Aldrich, MABT401,1:500) and Mmp-13 (Invitrogen, PA5-33940, 1:100).

## Statistical analysis

All data were expressed as means ± standard deviation of three representative experiments. The normal distribution analysis was checked by the Shapiro-Wilk test and quantile-quantile. The homogeneity of variances was tested by the Brown-Forsythe test. Parametric statistics were conducted using GraphPad Prism software (GraphPad Software Inc.) by Student's t-test (unpaired and two-tailed) and one-way ANOVA, followed by Tukey post hoc test. The values were considered significantly different at $p < 0.05$.

## Reporting summary

Further information on research design is available in the Nature Portfolio Reporting Summary linked to this article.

## Data availability

Raw sequencing data generated in this study have been deposited in the NCBI SRA database under accession number "PRJNA1055605". Genome Database (OryCun2.0_NCBI, https://www.ncbi.nlm.nih.gov/genome) and GO Database (http://geneontology.org/) are used in this article. All other data supporting the findings of this study are available within the article and its supplementary files. Any additional requests for information can be directed to, and will be fulfilled by, the corresponding author. Source data are provided with this paper.

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

## Acknowledgements

The authors would like to thank Guolong Meng, Lingzhu Yu, and Jiao Lu (National Engineering Research Center for Biomaterials, Sichuan University) for helping in characterizing SEM and characterizing CLSM. This work was supported by the National Natural Science Foundation of China (Nos. 32071352 and 32271419) and the National Key R&D Project of China (No. 2018YFC1105900).

## Author contributions

C.K.Z. and X.L.: Conceptualization, Data curation, Formal analysis, methodology, writing-original draft, review, and editing. X.W.H. and Z.L.L. provided assistance in material preparation and animal studies. S.Q.B., W.N.Z., M.M.D., and J.L. helped to edit the manuscript. Q.J.: Funding acquisition, Resources. Z.K.Z.: Resources. Y.J.F.: Funding acquisition, Resources, Supervision. X.D.Z.: Resources and project administration. Y.S.: Conceptualization, writing-review and editing, Funding acquisition, and Resources.

## Competing interests

The authors declare no competing interests.
