## [Peer Review File · Nature Communications]

REVIEWER COMMENTS

Reviewer #1 (Remarks to the Author):

I have read with great interest the review entitled 'Molecular co-assembled strategy tuning protein conformation for cartilage regeneration' by Chengkun Zhao et al.

The authors report on the development of a molecular Velcro inspired peptide and gelatin co-assembly strategy, in which amphiphilic supramolecular tripeptides were attached to the molecular chain of gelatin methacryloyl via intra- and intermolecular interactions. They report on the highly ordered β -sheet content of the co-assembled hydrogels, their improved compressive strength and the promoted cell proliferation and chondrogenic differentiation via transcriptome analysis. Reduced inflammatory responses have been described following subcutaneous implantation in mice, while hydrogel fixation in rabbit knee joint defects by photocuring has been found to lead to hyaline cartilage regeneration after three months.

The methodology is sound.

The results have been properly discussed and support the conclusions.

This is a very well-written and clearly presented manuscript.

I have some comments :

- Please discuss the role of the pore size of the produced scaffolds on cell infiltration towards the cartilage regeneration. The SEM images of Figure 1 h display pores with a diameter of ~ 10 - $20 \mu\text{m}$. Are pores of this size optimal for chondrogenesis? Please elaborate on this.
- Please elaborate on the effect of photocuring on the BM-MSCs of the cell-laden hydrogels.

And a few minor comments:

- In Fig. S3, y axis, please correct 'Storage'
- In Fig. S4, in the legend they refer to 'wavenumber range of 400 - 4000 cm^{-1} ', but the presented spectra indicate a range of 600 - 3700 cm^{-1} . Please adjust

Reviewer #2 (Remarks to the Author):

Dear authors,

your study is well written and interesting for the field of tissue engineering.

However, there are some critical points to address as follow:

1. Line 59-60: The authors describe gelatin as a molecule with similar amino acid sequences to collagen and a lower level of immunogenicity. Nevertheless, we know that collagen has a precise tridimensional shape made of a central domain and two portions called telopeptides, which are thought to be responsible for its immunogenicity. For this reason, telopeptides are often cleaved off enzymatically when used both in animal models and humans. On the other hand, telopeptides are important for flow crystallization, self-assembly, and cross-linking of collagen. What about gelatin's tridimensional shape? Please specify.

- Lynn AK, Yannas IV, Bonfield W. Antigenicity and immunogenicity of collagen. *J Biomed Mater Res B Appl Biomater* 2004;71(2):343-54. doi:10.1002/jbm.b.30096

- Suh DS, Lee JK, Yoo JC, Woo SH, Kim GR, Kim JW, et al. Atelocollagen Enhances the Healing of Rotator Cuff Tendon in Rabbit Model. *Am J Sports Med* 2017;45(9):2019-27. doi:10.1177/0363546517703336

- Paten JA, Siadat SM, Susilo ME, Ismail EN, Stoner JL, Rothstein JP, et al. Flow-Induced Crystallization of Collagen: A Potentially Critical Mechanism in Early Tissue Formation. *ACS Nano* 2016;10(5):5027-40. doi:10.1021/acsnano.5b07756

2. Every animal study should follow the ARRIVE 2.0 guidelines. These ten items are the minimum that must be included in any manuscript describing animal research. This information is necessary for readers and reviewers to assess the reliability of the findings. Please have these guidelines in your manuscript.

- Percie du Sert N, Ahluwalia A, Alam S, Avey MT, Baker M, Browne WJ, Clark A, Cuthill IC, Dirnagl U, Emerson M, Garner P, Holgate ST, Howells DW, Hurst V, Karp NA, Lazic SE, Lidster K, MacCallum CJ, Macleod M, Pearl EJ, Petersen OH, Rawle F, Reynolds P, Rooney K, Sena ES, Silberberg SD, Steckler T, Würbel H. Reporting animal research: Explanation and elaboration for the ARRIVE guidelines 2.0. *PLoS Biol.* 2020 Jul 14;18(7):e3000411. doi: 10.1371/journal.pbio.3000411. PMID: 32663221; PMCID: PMC7360025.

3. Line 389-390: More description of the surgical procedure needs to be done. The authors describe the diameter of the defect but not the depth and how they perform the defect (Do they reach the subchondral bone and see any bleeding?) Is this a chondral defect or an osteochondral defect? What about randomization of the defects? Please clarify and describe better the surgical procedure.

4. Line 391-392: Animals were grouped as follows: GelMA, Gel-co-MGFF, and normal (without surgery). It has been demonstrated that rabbits are a good model for cartilage defects. Nonetheless, the ability of self-healing of cartilage tissue in this animal model is very different to humans, and it is able to form a better cartilage when a hole is created. We believe there is a lack of a control group and authors should consider adding a new group as “negative control group” where a similar hole is created on the cartilage without performing any substance injection.

- Chu CR, Szczodry M, Bruno S. Animal models for cartilage regeneration and repair. *Tissue Eng Part B Rev.* 2010 Feb;16(1):105-15. doi: 10.1089/ten.TEB.2009.0452. PMID: 19831641; PMCID: PMC3121784.
- Wei, X., Gao, J., and Messner, K. Maturation-dependent re- pair of untreated osteochondral defects in the rabbit knee joint. *J Biomed Mater Res* 34, 63, 1997.
- Dell’Accio, F., Vanlauwe, J., Bellemans, J., Neys, J., De Bari, C., and Luyten, F.P. Expanded phenotypically stable chondrocytes persist in the repair tissue and contribute to cartilage matrix formation and structural integration in a goat model of autologous chondrocyte implantation. *J Orthop Res* 21, 123, 2003.

5. Line 397-399: Have the authors analyzed how the data is distributed? Have they used the Shapiro-wilk test or similar to discriminate between parametric and non-parametric statistics? Please specify

Reviewer #3 (Remarks to the Author):

Dear referees,

Thank you very much for your attention on our manuscript.

We highly appreciate the valuable comments and suggestions of reviewers, which greatly helped us to improve the quality of our manuscript. According to these comments and suggestions, we revised the manuscript carefully.

We hereby submit our revised manuscript entitled “***Molecular co-assembled strategy tuning protein conformation for cartilage regeneration***” for your consideration. The point-to-point response to the reviewers’ comments was listed as follows:

Response to Reviewer 1

Comment 1: *Please discuss the role of the pore size of the produced scaffolds on cell infiltration towards the cartilage regeneration. The SEM images of Figure 1 h display pores with a diameter of ~10-20 μm . Are pores of this size optimal for chondrogenesis? Please elaborate on this.*

Reply: Thanks for your valuable suggestion. Cell infiltration is a key behavior of cartilage tissue engineering because it is the basis of cell aggregation and differentiation¹. Therefore, it is necessary to study the cell infiltration towards cartilage regeneration. Generally speaking, scaffolds with smaller pore sizes could restrain cell proliferation and infiltration, while the larger pore size was beneficial to cell infiltration². However, in our research, BMSCs were 3D encapsulated into hydrogels, and cells were located inside rather than on the surface of the hydrogel.

Literature reported that the small pore size (60-125 μm) was an important factor in BMSC aggregation and subsequent chondrogenic differentiation, but the larger pore size (425-600 μm) was critical for vascularization and subsequent bone formation^{1,2}. As shown in SEM images (Fig. 1h), small pore sizes (around 20 μm) were exhibited in four groups, which is close to the reported small pore size (60-125 μm) for facilitating cartilage regeneration². This might be one of the potential reasons for the maintenance of chondrocyte phenotype *in vitro* and cartilage regeneration *in vivo*.

In addition, this scaffold with a pore diameter of 60-125 μm prevented endochondral ossification, while still allowing for cell migration and cartilage formation². However, it is not sure whether the pores with a diameter of ~10-20 μm are optimal for chondrogenesis, and the optimal pore size for cartilage regeneration still needs to be explored systematically. All these discussions about the effect of pore size on cell infiltration and chondrogenesis have been supplemented in the discussion part of the revised manuscript, which has been highlighted in red.

1. Wu, J. & Hong, Y. Enhancing cell infiltration of electrospun fibrous scaffolds in tissue regeneration. *Bioactive Materials* **1**, 56-64 (2016).

2. Gupte, M.J. et al. Pore size directs bone marrow stromal cell fate and tissue regeneration in nanofibrous macroporous scaffolds by mediating vascularization. *Acta Biomaterialia* **82**, 1-11 (2018).

Comment 2: *Please elaborate on the effect of photocuring on the BM-MSCs of the cell-laden hydrogels.*

Reply: Thanks for your valuable suggestion. The effect of ultraviolet radiations on cells has previously been reported, such as DNA damage through the formation of cyclobutane pyrimidine

dimer (CPD)³. However, the side effects were related to the irradiation doses and wavelengths of UV radiation. The longer wavelength and lower dose ensured the higher cell viability³. Meanwhile, the potential of blue light and LAP for photo-encapsulation of living cells has also been explored, which suggested that the blue light with 385 nm and 405 nm would hardly influence the cell viability (exceed 95%)⁴.

In this study, the irradiation of 405 nm blue light induced LAP to form radicals, which would initiate the free radical polymerization of GelMA. We evaluated the effect of photo-curing procedure on the BMSCs through live/dead staining and CCK-8. As shown in Fig. S5, the introduction of LAP initiator (BMSCs + LAP and BMSCs + LAP + Light group) could impede cell proliferation. However, the blue light (405 nm) could not affect cell viability and proliferation. The formed radicals of LAP might be the key reason for inhibiting cell proliferation. The related discussion was added in the result part of the revised manuscript and highlighted in red.

3. Masuma, R., Kashima, S., Kurasaki, M. & Okuno, T. Effects of UV wavelength on cell damages caused by UV irradiation in PC12 cells. *Journal of Photochemistry and Photobiology B: Biology* 125, 202-208 (2013).

4. Fairbanks BD, Schwartz MP, Bowman CN & Anseth KS. Photoinitiated polymerization of PEG-diacrylate with lithium phenyl-2,4,6-trimethylbenzoylphosphinate: polymerization rate and cytocompatibility. *Biomaterials* 30(35), 6702-7 (2009).

Fig. S5. The effect of photocuring on BMSCs. (a) Live/dead images of BMSCs under different conditions. The four groups were BMSCs (without treatment), BMSCs + light (UV irradiation), BMSCs + LAP (0.5% LAP in medium, w/v), and BMSCs + LAP + Light (0.5% LAP in medium and initiated by UV light, w/v). (b) CCK-8 analysis for BMSCs in four groups.

Comment 3: In Fig. S3, y axis, please correct 'Storage'.

Reply: Thanks for your reminder. The ‘Storgae’ has been replaced by the word ‘Storage’ in Fig. S3, which was highlighted in red.

Comment 4: *In Fig. S4, in the legend they refer to ‘wavenumber range of 400-4000 cm⁻¹, but the presented spectra indicate a range of 600-3700 cm⁻¹. Please adjust.*

Reply: Thanks for your reminder. Since the presented spectra indicated a range of 600-3700 cm⁻¹ in Fig. S4, we have corrected it into ‘These spectra were detected by Fourier transform infrared spectroscopy (Nicolet 6700, USA) with a 2 cm⁻¹ resolution and presented in the wavenumber range of 600-900 cm⁻¹, 1620-1740 cm⁻¹, and 2600-3700 cm⁻¹’.

Response to Reviewer 2

Comment 1: *Line 59-60: The authors describe gelatin as a molecule with similar amino acid sequences to collagen and a lower level of immunogenicity. Nevertheless, we know that collagen has a precise tridimensional shape made of a central domain and two portions called telopeptides, which are thought to be responsible for its immunogenicity. For this reason, telopeptides are often cleaved off enzymatically when used both in animal models and humans. On the other hand, telopeptides are important for flow crystallization, self-assembly, and cross-linking of collagen. What about gelatin's tridimensional shape? Please specify.*

Reply: We thank the reviewer for these important comments. As you mentioned above, telopeptides are significant in the self-assembly and tridimensional shape of collagen⁵⁻⁷. However, they also cause immunogenicity, which is the reason for the telopeptides removal enzymatically used in biomedical application^{8,9}. Here, we described gelatin as a molecule with similar amino acid sequences to collagen and a lower level of immunogenicity in this manuscript. According to the helpful comments from this reviewer, we corrected the inappropriate description as follow:

Gelatin is a collagen derivative with a lower level of immunogenicity, which was on account of the destruction of immunogenic structure of native collagens¹⁰⁻¹².

It is reported that gelatin is a heterogeneous mixture of peptides derived from the parent protein collagen by procedures involving the destruction of cross-linkages among the polypeptide chains along with some breakage of polypeptide bonds¹³. Some destruction of cross-linkages among the polypeptide chains results in some damage of triple helix structure^{13,14}. The chemical structure of gelatin consists of different polypeptide chains, such as of α -chains (one polymer/single chain), β -chains (two α -chains covalently crosslinked), and γ -chains (three covalently crosslinked α -chains) with a molar mass of around 90×10^3 , 180×10^3 and 300×10^3 g/mol, respectively¹⁵, which presented the significant difference with collagen triple helix structure. Compared with collagen, the relatively lower immunogenicity of gelatin was probably because some telopeptides were degraded enzymatically and the triple helix structure was denatured¹⁰⁻¹².

5. Lynn, A.K., Yannas, I.V. & Bonfield, W. Antigenicity and immunogenicity of collagen. *J. Biomed. Mater. Res. Part B Appl. Biomater.* **71B**, 343-354 (2004).

6. Shayegan, M., Altindal, T., Kiefl, E. & Forde, N.R. Intact telopeptides enhance interactions between collagens. *Biophys J* **111**, 2404-2416 (2016).

7. Hong, H. et al. Removing cross-linked telopeptides enhances the production of low-molecular-weight

collagen peptides from spent hens. *J. Agric. Food Chem.* **65**, 7491-7499 (2017).

8. Suh, D.-S. et al. Atelocollagen enhances the healing of rotator cuff tendon in rabbit model. *Am. J. Sports Med.* **45**, 2019-2027 (2017).

9. Paten, J.A. et al. Flow-induced crystallization of collagen: A potentially critical mechanism in early tissue formation. *ACS Nano* **10**, 5027-5040 (2016).

10. Zheng, M. et al. Skin-inspired gelatin-based flexible bio-electronic hydrogel for wound healing promotion and motion sensing. *Biomaterials* **276**, 121026 (2021).

11. Dong, Y. et al. Injectable and tunable gelatin hydrogels enhance stem cell retention and improve cutaneous wound healing. *Adv. Funct. Mater.* **27**, 1606619 (2017).

12. Sharifi, S. et al. Tuning gelatin-based hydrogel towards bioadhesive ocular tissue engineering applications. *Bioact. Mater.* **6**, 3947-3961 (2021).

13. Liu, D., Nikoo, M., Boran, G., Zhou, P. & Regenstein, J.M. Collagen and gelatin. *Annu Rev Food Sci Technol* **6**, 527-557 (2015).

14. Alipal, J. et al. A review of gelatin: Properties, sources, process, applications, and commercialisation. *Mater. Today: Proc.* **42**, 240-250 (2021).

15. Mariod, A. A., & Fadul, H. Gelatin, source, extraction and industrial applications. *Acta Scientiarum Polonorum Technologia Alimentaria* **12(2)**, 135-147 (2013).

Comment 2: *Every animal study should follow the ARRIVE 2.0 guidelines. These ten items are the minimum that must be included in any manuscript describing animal research. This information is necessary for readers and reviewers to assess the reliability of the findings. Please have these guidelines in your manuscript.*

Reply: Thanks for your valuable suggestion. According to the helpful comments, we have made a supplement in the methods and results part of the revised manuscript following the ARRIVE 2.0 guidelines, which were highlighted in red. The descriptions are as follows:

Seven adult New Zealand white rabbits (2.5-3.0 kg, male) were adopted and randomly divided into four groups (Blank, GelMA, Gel-co-MGFF, and Normal) for *in vivo* evaluation of cartilage reconstruction in this research. After being anesthetized, the experimental rabbits were fixed on the operating table in a supine position. The legs around the knee joints were shaved, disinfected, and draped. After an incision (about 1.5 cm long) was made on the inner side of the knee joint, the patella was everted to expose the trochlear groove. Then, a sterile hollow metal tube was applied to drill a circle (3 mm in diameter) on the center of the trochlear groove, and the hyaline cartilage was scraped out with a scalpel until the calcified cartilage layer was exposed. The animals were included in the study if they underwent successful hyaline cartilage removal, while the animals would be excluded if the subchondral bone was damaged and/or there was blood leaking during this process. For GelMA and Gel-co-MGFF groups, precursor solutions were injected into the defective area on the trochlear groove of each side and were cross-linked with initiating by LAP under the irradiation of blue light ($\lambda = 405$ nm). The defective area of rabbits in blank groups was treated without filler. All surgically treated rabbits were sutured and disinfected with iodine after patella repositioning. Surgical operation was not applied on the normal group rabbits. Each group has three duplicate samples. After euthanasia, all samples were taken out after 3 months, then the knee appearance was photographed. The macroscopic and histological results were analyzed by three investigators who were blind to the groups. Five different researchers were randomly involved in animal group distribution, operation, and evaluation.

Comment 3: *Line 389-390: More description of the surgical procedure needs to be done. The authors describe the diameter of the defect but not the depth and how they perform the defect (Do they reach the subchondral bone and see any bleeding?) Is this a chondral defect or an osteochondral defect? What about randomization of the defects? Please clarify and describe better the surgical procedure.*

Reply: Thanks for your valuable suggestion. According to the helpful comments, we have made a supplement in the methods part of the revised manuscript following the ARRIVE 2.0 guidelines to complete the surgical procedure. Our surgical procedure was conducted as follows:

Seven adult New Zealand white rabbits (2.5-3.0 kg, male) were adopted and randomly divided into four groups (Blank, GelMA, Gel-co-MGFF, and Normal) for *in vivo* evaluation of cartilage reconstruction in this research. After being anesthetized, the experimental rabbits were fixed on the operating table in a supine position. The legs around the knee joints were shaved, disinfected, and draped. After an incision (about 1.5 cm long) was made on the inner side of the knee joint, the patella was everted to expose the trochlear groove. Then, a sterile hollow metal tube was applied to drill a circle (3 mm in diameter) on the center of the trochlear groove, and the hyaline cartilage was scraped out with a scalpel until the calcified cartilage layer was exposed. The animals were included in the study if they underwent successful hyaline cartilage removal, while the animals would be excluded if the subchondral bone was damaged and/or there was blood leaking during this process. For GelMA and Gel-co-MGFF groups, precursor solutions were injected into the defective area on the trochlear groove of each side and were cross-linked with initiating by LAP under the irradiation of blue light ($\lambda = 405$ nm). The defective area of rabbits in blank groups was treated without filler. All surgically treated rabbits were sutured and disinfected with iodine after patella repositioning. Surgical operation was not applied on the normal group rabbits. Each group has three duplicate samples. After euthanasia, all samples were taken out after 3 months, then the knee appearance was photographed. The macroscopic and histological results were analyzed by three investigators who were blind to the groups. Five different researchers were randomly involved in animal group distribution, operation, and evaluation.

These descriptions of surgical procedure details have been expanded in the revised manuscript and highlighted in red.

Comment 4: *Line 391-392: Animals were grouped as follows: GelMA, Gel-co-MGFF, and normal (without surgery). It has been demonstrated that rabbits are a good model for cartilage defects. Nonetheless, the ability of self-healing of cartilage tissue in this animal model is very different to humans, and it is able to form a better cartilage when a hole is created. We believe there is a lack of a control group and authors should consider adding a new group as “negative control group” where a similar hole is created on the cartilage without performing any substance injection.*

Reply: Thank you for the valuable advice. We agreed with the reviewer's opinion on adding a negative control group. The reasons for the lack of a negative control group are as follows:

In this study, we actually conducted a blank group for the experimental study. Considering that it has been proved in many literatures that under defect conditions, hyaline cartilage regeneration of blank group could not be achieved, only part of the fiber tissue was filled¹⁶⁻¹⁸. Therefore, we did not integrate relevant data into this manuscript. We apologize for any inconvenience it may have caused

you.

16. Chen, Y. R. et al. Kartogenin-conjugated double-network hydrogel combined with stem cell transplantation and tracing for cartilage repair. *Adv. Sci.* **9**, 2105571 (2022).

Representative (a) Gross view (b) MRI (c) H&E staining (d) TB staining (e) Col II IHC (f) SEM images of the blank groups.

17. Li, P. et al. Chitosan hydrogel/3D-printed poly(ϵ -caprolactone) hybrid scaffold containing synovial mesenchymal stem cells for cartilage regeneration based on tetrahedral framework nucleic acid recruitment. *Biomaterials* **278**, 121131 (2021).

Representative (a) Gross view (b) MRI (c) H&E staining (d) SO-FG staining (e) Sirius-red staining (f) Col II IHC images of the blank groups.

18. Cai, H. et al. BMSCs-assisted injectable Col I hydrogel-regenerated cartilage defect by reconstructing superficial and calcified cartilage. *Regen. Biomater.* **7**, 35-45 (2020). (One of our previous works)

Representative (a) Gross view (b) SO-FG staining (c) TB staining (d) H&E staining (e) Sirius-red staining images of the blank groups.

Now, we supplemented relevant data of this blank group with a similar hole as “negative control group”. As shown in Fig. S9, the histological staining of sections demonstrated ineffective

cartilage regeneration induced by the blank group. The related discussion was presented in the result part of the revised manuscript and highlighted in red.

Fig. S9. (a) Schematic diagram of BMSCs-laden hydrogels injected into the rabbit knee defective area ($\Phi = 3$ mm). (b, c) The process of injection *in situ* and light irradiation. (d1-f4) Representative pictures of Sirius-red, Masson's trichrome, and Safranin O-Fast green staining after 3 months of implantation. The arrows point to the defective area.

Comment 5: Line 397-399: Have the authors analyzed how the data is distributed? Have they used the Shapiro-wilk test or similar to discriminate between parametric and non-parametric statistics? Please specify.

Reply: Thanks for your valuable suggestion. The normality distribution test has limitations on the number of data points. For the sufficient samples, such as Fig. 1n, Fig. 6d, and Fig. 6f, the normal distribution analysis was checked by the Shapiro-Wilk test and quantile-quantile. The homogeneity of variances was checked by the Brown-Forsythe test. Parametric statistics was used in data analysis which conforms to the normal distribution and homogeneity test of variance.

Response to Reviewer 3

Comment 1: I co-reviewed this manuscript with one of the reviewers who provided the listed reports. This is part of the Nature Communications initiative to facilitate training in peer review

and to provide appropriate recognition for Early Career Researchers who co-review manuscripts.

Reply: Thank you for your review of our manuscript. The point-to-point response to the comments has been shown in the response to reviewers.

We would like to thank the reviewers again for taking the time to review our manuscript and expect your comments on this revised manuscript at your convenience.

Thanks and all the best.

Yours sincerely,

Prof. Dr. Yong Sun

National Engineering Research Center for Biomaterials

Sichuan University

Chengdu, China

REVIEWERS' COMMENTS

Reviewer #1 (Remarks to the Author):

The revised manuscript version has adequately addressed the comments raised by this reviewer and maybe suitable for publication in Nature Communications.

Reviewer #2 (Remarks to the Author):

The article is well written and clear. The authors answered to our comments clearly and point to point. We do not have any edit or comment.

Reviewer #3 (Remarks to the Author):

Dear authors,

I had the chance to co-review this manuscript.

Thank you very much for your revisions that greatly improved the manuscript.

To us no additional changes are required.